# Bile Acid and Fibroblast Growth Factor 19 Regulation in Obese Diabetics, and Non-Alcoholic Fatty Liver Disease after Sleeve Gastrectomy

**DOI:** 10.3390/jcm8060815

**Published:** 2019-06-07

**Authors:** Hsien-Hao Huang, Wei-Jei Lee, Shu-Chun Chen, Tung-Fang Chen, Shou-Dong Lee, Chih-Yen Chen

**Affiliations:** 1Department of Emergency Medicine, Taipei Veterans General Hospital, Taipei 11217, Taiwan; hhhuang@vghtpe.gov.tw; 2Institute of Emergency and Critical Medicine, National Yang-Ming University School of Medicine, Taipei 11221, Taiwan; 3Department of Surgery, Min-Sheng General Hospital, Taoyuan 33044, Taiwan; wjlee_obessurg_tw@yahoo.com.tw; 4Taiwan Society for Metabolic and Bariatric Surgery, Taipei 11031, Taiwan; N002916@e-ms.com.tw; 5Department of Nursing, Chang‐Gung Institute of Technology, Taoyuan 33303, Taiwan; 6Medical Affairs Office, Taipei City Hospital Yangming Branch, Taipei 11146, Taiwan; Z4430@tpech.gov.tw; 7Division of Gastroenterology, Department of Internal Medicine, Cheng-Hsin General Hospital, Taipei 11220, Taiwan; 8Division of Gastroenterology and Hepatology, Department of Medicine, Taipei Veterans General Hospital, Taipei 11217, Taiwan; 9Faculty of Medicine, National Yang-Ming University School of Medicine, Taipei 11221, Taiwan; 10Chinese Taipei Society for the Study of Obesity, Taipei 11031, Taiwan

**Keywords:** diabetes mellitus, FGF 19, non-alcoholic fatty liver disease, sleeve gastrectomy, total bile acid

## Abstract

Background: Sleeve gastrectomy (SG) is an effective treatment for obesity and type 2 diabetes mellitus (T2DM), and non-alcoholic fatty liver disease (NAFLD); however, the mechanism is not completely understood. Bile acids and fibroblast growth factors (FGFs) are involved in the regulation of energy metabolism. Methods: We investigated the roles of total bile acid and FGF 19 in T2DM remission and NAFLD improvement in obese subjects undergoing SG. A total of 18 patients with obesity and T2DM undergoing laparoscopic SG were enrolled in this study. Serial plasma total bile acid and FGF 19 levels were measured, while the fatty liver index was calculated before and after surgery. Results: The FGF 19 level significantly increased, and the total bile acid level and fatty liver index decreased 1 year after surgery. The complete T2DM remission rate was 66.7% one year after surgery; the complete remitters had significantly lower FGF 19 levels and higher insulin levels than the non-complete remitters. The complete remitters also had significantly decreased total bile acid levels and increased FGF 19 levels 1 year after surgery compared with those before surgery. The fatty improvers had significantly decreased total bile acid levels and increased FGF 19 levels 1 year after surgery compared with those before surgery. Conclusion: The total bile acids level and fatty liver index decreased, and the FGF 19 levels increased 1 year after SG. Both T2DM complete remitters and NAFLD improvers showed significantly decreased total bile acid levels and increased FGF 19 levels 1 year after SG. Plasma total bile acids and FGF 19 might have roles in T2DM remission and NAFLD improvement. Low preoperative FGF 19 levels might be a predictor for NAFLD improvement after SG.

## 1. Introduction

Bile acids are derived from cholesterol in the liver and modulate hepatic lipid, glucose, and energy metabolism through nuclear and membrane receptors [1,2,3]. They are the ligands for the farnesoid X receptor (FXR) and can activate it in pancreatic β-cells to increase insulin secretion [4]. Circulating bile acids was correlated with insulin sensitivity [5]. Bile acids can also activate the FXR and Takeda G-protein receptor 5 in the liver and intestines and consequently diversely affect metabolism and inflammation [1,6]. By binding to the Takeda G-protein receptor 5, circulating bile acids activate glucagon-like peptide-1 release to improve insulin secretion and sensitivity [7]. Therefore, acting on the FXR might be a potential method to improve insulin and glucose sensitivity in diabetes mellitus (DM) and non-alcoholic fatty liver disease (NAFLD) [2,6]. Fexaramine, an intestinal FXR agonist, was shown to improve lipid profiles, increase GLP-1 secretion, and improve glucose and insulin tolerance [8]. Inconsistently, glycine-β-muricholic acid, as a selective high-affinity FXR inhibitor, inhibits FXR signaling and improves obesity, insulin resistance, and hepatic steatosis in mouse models of obesity [9]. Thus, more investigations are needed to clarify the relationships of bile acids between DM and NAFLD.

Fibroblast growth factor (FGF) 19 is produced in response to activation of the transcription factor FXR by bile acids [10]. The human FGF family has at least 22 members. FGF 19 is produced and secreted into the portal circulation, a pronounced diurnal rhythm with peaks occurring 90–120 min after the postprandial rise in serum bile acid levels [11]. The binding of FGF 19 to the hepatocyte cell surface FGF receptor 4 leads to reduction of hepatic bile salt synthesis [11], negative feedback loop, and gallbladder refilling [10,12]. Emerging evidence reveals that the serum FGF 19 levels are decreased in patients with type 2 DM (T2DM) [13]. Altering the flow of endogenous bile can result in FGF response changes after bariatric surgery. Although increased FGF 19 levels have been observed in patients with morbid obesity but without DM at 12 months after sleeve gastrectomy (SG) [14], information regarding bile acids, FGF 19 and NAFLD dynamics in patients with T2DM after bariatric surgery is essential, but remains completely lacking.

SG, a bariatric surgery, is currently considered one of the effective treatments for patients with morbid obesity (body mass index (BMI), > 35 kg/m^2^) [15] and Asian patients with non-morbid obesity (BMI, < 35 kg/m^2^) and not-well-controlled T2DM [16]. Moreover, SG induces up to 50% and 84.6% of T2DM remission rate 1 year after surgery in patients with non-morbid obesity [16] and morbid obesity [17]. Although bariatric surgery yields valid and durable outcomes, the mechanisms by which it causes weight loss and T2DM and NAFLD resolutions are not well elucidated.

Bile acids and FGF 19 may explain the pleotrophic metabolic effects observed after bariatric surgery. They could serve as therapeutic targets for novel surgical refinement or pharmaceuticals. In this study, we prospectively investigated the serial changes in the bile acids level, FGF 19 level, and NAFLD status before and after SG. Further, we determined the relationship between FGF 19 and improvement of either T2DM or NAFLD because of their close interactions. We hope to elucidate the roles of bile acids and FGF 19 and offer mechanistic insights into human DM remission and fatty liver improvement.

## 2. Materials and Methods

### 2.1. Study Design and Patients

A hospital-based design was adapted in the present study. This study was conducted at the Department of Surgery of the Min-Sheng General Hospital and at the Taipei Veterans General Hospital, and was approved by the Ethics Committee of each hospital (approval number: MSIRB2015020). Patients with T2DM who received SG were enrolled in the present study. The inclusion criteria were as follows: (1) T2DM onset more than 6 months with glycated hemoglobin levels of ≥ 8% and intensive medical care under an endocrinologist, (2) BMI of ≥ 25 kg/m^2^, (3) willingness to receive accessory therapy with diet control and exercise, (4) willingness to be followed up, and (5) willingness to sign the informed consent.

The exclusion criteria were as follows: (1) cancer diagnosed in recent 5 years, (2) human immunodeficiency viral infection or active pulmonary tuberculosis, (3) cardiovascular diseases or cardiovascular instability in the recent 6 months, (4) pulmonary embolism or uncontrolled coagulative diseases, (5) serum creatinine levels of > 2 mg/dL, (6) chronic hepatitis B or C, liver cirrhosis, or inflammatory bowel diseases, (7) acromegaly or history of other organ transplantation, (8) history of bariatric surgery or gastrointestinal surgery other than cholecystectomy or history of abdominal septicemia, (9) alcohol or drug abuse or psychiatric diseases, and (10) other conditions that could result in patients’ uncooperation.

Eighteen patients who received laparoscopic SG were enrolled into this prospective, longitudinal study. Treatment decision was based on the clinical background of the individual subjects. After SG, all patients received the clinics follow-up and consultation of dieticians. All were encouraged life style modifications, such as walking 10,000 steps/per day and strict diet control.

### 2.2. Laparoscopic SG

For laparoscopic SG, the greater curvature, including the complete fundus, was resected from the distal antrum (4 cm proximal to the pylorus) to the angle of His. A standard 5-trocar laparoscopic technique was applied using a laparoscopic stapler (EndoGIA; Tyco, United States Surgical Corporation, Norwalk, CT, USA) with a 60-mm cartridge (3.5-mm stapler height, blue load) to divide the stomach ~2 cm wide along the lesser curvature side. The resected portion of the stomach was extracted from the extended peri-umbilical trocar site. A running absorbable suture was applied to the stapler line to prevent hemorrhage and leakage. No drain tube was left [16].

### 2.3. Blood Sampling

Blood samples were obtained from the antecubital vein between 8 AM and 11 AM after an overnight fast before metabolic surgery (M0). The samples were immediately transferred into a chilled glass tube containing disodium EDTA (1 mg/mL) and aprotinin (500 units/mL) and stored on ice during collection; further, they were centrifuged, plasma-separated, aliquoted into polypropylene tubes, and stored at −20°C for later analysis. The aliquots were number-coded. Blood sampling was repeated immediately 3 months (M3) and 1 year (M12) after SG between 9 AM and 11 AM after an overnight fast. The levels of plasma FGF 19 and serum total bile acid were measured in the blood samples.

### 2.4. Definition of DM Remission and Insulin Resistance

Remission of T2DM was defined as fasting glucose levels of 100–125 mg/dL with glycated hemoglobin levels of < 6.5% in the absence of pharmacotherapy, including oral hypoglycemics and insulin [18]. Insulin resistance was assessed using the homeostatic model assessment-index, calculated as follows: plasma glucose level (mmol/L) × insulin level (μU/mL)/22.5. β-cell function was assessed using the homeostatic model assessment index-β (HOMA-β) index, calculated as follows: {20 × insulin level (μU/mL)/(plasma glucose level (mmol/L) − 3.5) [19].

### 2.5. Definition of NAFLD based on the Fatty Liver Index (FLI)

NAFLD has a feature of metabolic syndromes. The FLI can be non-invasively assessed for predicting the presence of NAFLD and calculated as follows: (e ^0.953 × ln (triglyceride level) + 0.139 × BMI + 0.718 × ln (ggt) + 0.053 × waist circumference − 15.745^/(1+ e ^0.953 × ln (triglyceride level) + 0.139 × BMI + 0.718 × ln (ggt) + 0.053 × waist circumference − 15.745^) × 100 [20]. An FLI of ≥30 for determining NAFLD has a specificity of 69% and a positive likelihood ratio of 2.43 [21].

### 2.6. Measurement of the Plasma FGF 19 and Serum Total Bile Acid Level

The fasting blood samples obtained will be used to determine the plasma levels of FGF 19 and serum levels of total bile acids. Enzyme immuno-assays for plasma FGF 19 (R&D Systems, Minneapolis, MN, USA) was performed in duplicate [22]. Fasting total serum bile acids were assayed using the 3α-hydroxysteroid dehydrogenase method (Fumouze Diagnostics, Levallois-Perret, France); this was also performed in duplicate [23].

### 2.7. Statistical Analysis

All statistical analyses were performed using the Statistical Package for Social Sciences software version 12.01 (SPSS, Inc., Chicago, IL, USA). The chi-square test was used to compare categorical variables and the Mann-Whitney test to compare continuous variables. Friedman’s repeated measures analysis of variance on ranks was used to compare between the baseline and postoperative variables serially. Correlations between two groups were analyzed using Spearman’s correlation method. A *p* value of < 0.05 was considered statistically significant.

## 3. Results

A total of 18 patients with obesity and T2DM who underwent SG (7 men and 11 women) were enrolled. The baseline average age, body weight, and BMI were 39.5 ± 8.9 years old, 99.4 ± 19.0 kg, and 35.6 ± 5.1 kg/m^2^, respectively. The duration of T2DM was 3.1 ± 2.9 years. All enrolled patients underwent SG and were followed up for more than 1 year.

As shown in Table 1, all metabolic components, including body weight, BMI, waist circumference, and excess weight loss, significant improved after SG (*p* < 0.001). The parameters of DM, including fasting blood glucose, glycated hemoglobin, insulin, and C-peptide levels, and homeostatic model assessment-insulin resistance index, significantly improved 1 year after SG (*p* < 0.05). Only the HOMA-β did not significantly change. As for the hepatic enzymes 1 year after SG, the alanine aminotransferase and gamma glutamyl transferase levels significantly decreased (*p* < 0.05); however, the aspartate aminotransferase and alkaline phosphatase levels did not change. The total bile acid level significantly decreased from 16.8 ± 12.3 μM at M0 to 10.7 ± 8.2 μM at M3 and 7.3 ± 4.0 μM at M12 (Figure 1A, *p* < 0.05). As for the lipid profile after SG, the triglyceride levels decreased (*p* < 0.01), and the high-density lipoprotein cholesterol levels increased (*p* < 0.05); however, the total cholesterol and low-density lipoprotein cholesterol did not change. We also applied hepatic steatosis index [24] and lipid accumulation product [21,25] to validate our improvement of NAFLD one year after SG by FLI. SG successfully decreased the hepatic steatosis index and lipid accumulation product scores one year after SG (*p* < 0.001). As for NAFLD status, the FLI dramatically decreased 1 year after surgery (*p* < 0.001). As for the hepatokines, the FGF 19 levels significantly increased from 81.6 ± 59.3 pg/mL at M0 to 148.0 ± 129.7 pg/mL at M3 and 147.6 ± 62.0 pg/mL at M12 (Figure 1B, *p* < 0.05). The FGF 19 level had negative correlations with the C-peptide (Figure 2A, *p* = 0.029, Pearson correlation coefficient (ρ) = 0.583), alanine aminotransferase (Figure 2B, *p* = 0.014, Pearson correlation coefficient (ρ) = −0.581) and total bile acid levels (Figure 2C, *p* = 0.016, Pearson correlation coefficient (ρ) = 0.557) 1 year after surgery.

As shown in Table 2, the complete DM remission rate was 66.7% (12/18) 1 year after SG. Before SG, the medications for treating DM were oral anti-diabetic drugs in 16 patients (16/18, 88.9%), insulin in 2 patients (2/18, 11.1%), and statins in 8 (8/18, 44.4%) patients. After SG, only 1 (1/18, 5.6%) patient used oral anti-diabetic drugs 3 months after surgery, and 1 (1/18, 5.6%) patient used statins 1 year after surgery. None of the patients ever used glucagon-like peptide-1 agonist. In Table 2, there was no any statistical significant difference between complete remitters of DM (DM-CR group) and the non-complete remitters of DM (DM-non-CR group) in any variables before SG. One year after SG, the DM-CR group had significantly lower FGF 19 levels (126.3 ± 50.5 vs. 200.6 ± 46.9 pg/mL, *p* < 0.05), higher insulin levels (5.4 ± 2.7 vs. 2.4 ± 0.7 μU/mL, *p* < 0.05), lower glycated hemoglobin levels (5.5 ± 0.2 vs. 6.3 ± 0.4 %, *p* < 0.001), and higher HOMA-β index (110.8 ± 127.9 vs. 20.4 ± 15.0, *p* = 0.01) 1 year after SG than the DM-non-CR group. The DM-CR group had significantly lower total bile acid levels at M12 (Figure 3A, *p* < 0.01) and higher FGF 19 at M3 and M12 (Figure 3B, *p* < 0.05) than at M0; these findings were not observed in the DM-non CR group.

Based on defined FLI of ≥ 30 (Table 3), the fatty liver improvement rate was 69.2% (9/13) 1 year after SG surgery. Before SG, the improvers of fatty liver (FL-I group) demonstrated significantly lower body weight (94.4 ± 9.7 vs. 121.3 ± 19.7 kg, *p* < 0.05), lower waist circumferences (104.7 ± 6.1 vs. 122.9 ± 12.0 kg/m^2^, *p* < 0.05), lower FGF 19 (48.7 ± 18.0 vs. 101.2 ± 42.4%, *p* < 0.05) than the non-improvers of fatty liver (FL-non-I group). One year after SG, 5 patients were excluded due to missing data at 1 year after SG in the calculation of FLI (2 patients without triglyceride level, 2 patients without gamma glutamyl transferase, and 1 patient without waist circumference). The FL-I group demonstrated significantly lower body weight (69.0 ± 8.0 vs. 98.6 ± 10.8 kg, *p* < 0.01), lower BMI (24.8 ± 1.8 vs. 32.1 ± 3.1 kg/m^2^, *p* < 0.01), higher excess weight loss (78.7 ± 15.2 vs. 41.5 ± 11.5%, *p* < 0.01), lower triglyceride level (83.8 ± 24.9 vs. 123.3 ± 33.0 mg/dL, *p* = 0.05), higher high-density lipoprotein cholesterol level (55.9 ± 6.2 vs. 38.5 ± 5.3 mg/dL, *p* < 0.05), and HOMA-β index (42.6 ± 41.2 vs. 204.1 ± 180.2, *p* = 0.05) 1 year after SG than did the FL-non-I group. The FL-I group exhibited significantly lower total bile acid levels at M3 and M12 (Figure 4A, *p* < 0.05 and *p* < 0.05), higher FGF 19 levels at M12 (Figure 4B, *p* < 0.01), and lower gamma glutamyl transferase levels at M3 and M12 (Figure 4C, *p* < 0.05 and *p* < 0.05) than at M0; these findings were not observed in the FL-non-I group. Specifically, the FL-I group had significantly lower FGF 19 levels at M0 (Figure 4B) than the FL-non-I group. A receiver operating characteristic (ROC) curve analysis was performed to determine the potential role of FGF 19 level in the discrimination of FL-I group and FL-non-I group. The area under ROC curve between the two groups was 0.917 (95% confidence interval: 0.631-0.997; *p* < 0.0001). The optimal cut-off value for FGF 19 level between the two groups was 49.97 pg/mL. When FGF 19 was ≤ 49.97 pg/mL, the sensitivity, specificity, positive predictive value and negative predictive value for FL-I group were 77.8%, 100.0%, 100% and 66.7%, respectively.

## 4. Discussion

Our study investigated 18 patients with obesity and T2DM and showed that SG is an effective intervention to improve obesity (body weight, BMI, and waist circumference), insulin intolerance (fasting blood glucose, glycated hemoglobin, C-peptide, and insulin levels and homeostatic model assessment-insulin resistance index), liver functions (alanine aminotransferase and gamma glutamyl transferase levels), lipid profile (triglyceride and high-density lipoprotein cholesterol level) and NAFLD (FLI). Moreover, the total bile acid levels significantly decreased and FGF 19 levels significantly increased in the patients with obesity and T2DM 1 year after SG. In addition, the FGF 19 level had a negative correlation with the C-peptide, alanine aminotransferase and total bile acid levels 1 year after SG.

The DM-CR group had significant lower FGF 19 levels, higher insulin levels and higher HOMA-β index than the DM-non-CR group 1 year after surgery. The DM-CR group also had significantly lower total bile acid levels and higher FGF 19 levels at M12 than at MConversely, the FL-I group exhibited significantly lower FGF 19 levels before surgery and lower body weight, BMI, triglyceride levels, and high-density lipoprotein levels1 year after SG than did the FL-non-I group. Further, the FL-I group had significantly lower total bile acid levels, lower gamma glutamyl transferase levels, and FLI at M12 than at MBased on these results, total bile acid and FGF 19 might have some roles in DM remission and the fatty liver improvement one year after SG.

The FXR is a bile acid-binding receptor that regulates bile acid and triglyceride homeostasis. Without the FXR, the ability of SG to reduce body weight and improve glucose tolerance is substantially reduced [26]. In subjects with obesity, the levels of bile acids are low (3–4 μmol/L) [14,27,28,29]. Conversely, subjects with obesity and DM showed significantly higher fasting total bile acid levels than did those without DM (9.1 vs. 4.5 μmol/L, *p* < 0.05) [29]. This may be mainly attributed to the fact that individuals with DM exhibit low FXR expression [30]. Bile acid synthesis was upregulated, and the bile acid pool size increased in mice, which have no FXR [31]. According to the bile acid→FXR→ FGF 19→CYP7A1 pathway [32], free luminal bile acids could bind to the FXR and activate FGF 19 secretion. However, in DM, the activation of FGF 19 secretion is dampened owing to dysregulation of FXR expression. This view was confirmed in a finding that subjects with obesity and DM had lower serum levels of FGF 19 and higher levels of bile acid clinically [29], which is consistent with the results of our current study. We observed that the fasting baseline levels of total bile acids in patients with obesity and T2DM were ~16.8 μmol/L which is compatible with previous reports [29].

Previous studies on total bile acid levels after SG reported inconsistent findings; they may increase [28], not change [14,33] or decrease [34]. Dysregulation of the FXR in T2DM could be the confounding factor. Individuals with obesity and T2DM showed higher bile acid levels (~10.2 μmol/L [29]) than did those without DM (~3–4 μmol/L [14,27,28,29]). FXR-knockout mice have been shown to lose the abilities of reducing body weight and improving glucose tolerance after SG [26]. The results of our study found that high total bile acid, probably due to dysregulation of the FXR in T2DM, and subsequently low FGF 19 were detected before SG. After bariatric surgery, the FGF 19 level significantly increased in subjects with obesity [14,35] and subjects with obesity and DM [29]. Circulating FGF 19 was then proposed to bind to the receptors in the liver and suppress bile acid synthesis [10]. This feedback regulation of FGF 19 in human hepatic bile acid synthesis supports our new findings, which revealed decreasing bile acid levels 1 year after SG in the patients with obesity and T2DM, and the negative relationship between FGF 19 and total bile acid. Our study focused on patients with obesity and T2DM, and we found that SG significantly decreased the total bile acid level from 16.8 μM to 7.3 μM 1 year after surgery. Furthermore, the total bile acid level significantly decreased in the DM-CR group at 1 year after surgery compared with that at before surgery. Owing to the Y-Y paradox, i.e., identical BMI with different body fat in different ethnic populations [36], our previous study also showed that the 2-year rate of DM-CR was only 5.5% after SG, much less than 50% after gastric bypass [37]. The new findings in our study suggest the close relationship between total bile acid and DM remission after SG in our Asian population. The regulation of FXR might play an important role in subjects with obesity and DM. Using the bile acid sequestrant, colesevelam [38,39] or colestyramine [40], could also improve glycemic control in patients with T2DM. The FXR linking between bile acid and glucose metabolism is a potential pharmacological target for therapeutic applications.

Total bile acid had been proposed to improve in the pathogenesis of NAFLD and nonalcoholic steatohepatitis by inducing cytotoxicity [41]. A high prevalence (91%) of NAFLD and more than one-third (37%) of nonalcoholic steatohepatitis by histology were noted in patients with morbid obesity [42]. Bariatric surgery not only effectively induced weight loss, but also significantly improved the histological features of NAFLD in patients with morbid obesity after surgery [43,44,45]. The pathological features of nearly 85% of non-alcoholic steatohepatitis cases disappeared 1 year after bariatric surgery [46]. Our study found that the FL-I group had significantly decreased total bile acid levels at M3 and M12, as compared with at M0 after SG. These results suggest that total bile acid plays an important role in fatty liver improvement.

FGF 19 is a peptide with 216 amino acids and was proposed as a metabolic regulator in improving DM, hyperlipidemia, hepatic steatosis, and adiposity [22,47]. Obesity is an important factor associated with the FGF 19 levels [14,29]. The levels of FGF 19 in patients with obesity (BMI, range of 25–34.9 kg/m^2^) and morbid obesity (BMI, 35–66 kg/m^2^) were 151.8 pg/mL and 118.2 pg/mL, respectively [29]. The levels of FGF 19 have a negative correlation with BMI and patients with morbid obesity have lower FGF 19 levels than patients with obesity [13]. Further, patients with obesity and DM have significantly lower FGF 19 levels than patients with obesity but without DM [29].

As for the change in the FGF, 19 levels in patients with obesity after SG, the serum levels significantly increased at 6 months [14,33] and persisted for at least 2 years after surgery [14]. Our study further extended the abovementioned finding to patients with T2DM in that the FGF 19 level significantly increased in the patients with obesity and T2DM 1 year after SG. Furthermore, our study first showed that DM-CR group had significantly higher levels of FGF 19 and insulin than the DM-non-CR group 1 year after surgery. Taken together, FGF 19 plays a role in the remission of T2DM.

FGF 19 has an essential role in NAFLD. Liver specimens from patients with NAFLD had increased endoplasmic reticulum stress [48,49] and chronic endoplasmic reticulum stress has been linked to fatty liver disease [50] owing to increased FFA flux into the liver [51]. Lack of the FGF15/19 gene exacerbated endoplasmic reticulum stress and increased hepatic steatosis in response to a high-fat diet [50]. FGF 19 transgenic mice did not develop obesity and DM on a high-fat diet by reducing fat mass, increasing energy expenditure, increasing insulin sensitivity, and reducing hepatosteatosis [52]. Patients with NAFLD and insulin resistance had an impaired hepatic response to FGF19, leading to the dysregulation of lipid homeostasis [53]. Obesity showed an association between decreased FGF19 levels and NAFLD [54]. Our study corresponds to the above, and further investigates the FGF 19 and FLI score on NAFLD in patients with obesity and T2DM after SG. In different metabolic surgery, SG had a better capability to normalize liver function test than Roux-en-Y gastric bypass in patients with obesity and T2DM 1 year after surgery [55]. Non-invasive markers of liver fibrosis scores, including aspartate aminotransferase/alanine aminotransferase ratio, NAFLD fibrosis score, BARD score, and aspartate aminotransferase to platelet ratio index, have been used to evaluate the improvement of NAFLD after SG [56]. We also found that the FL-I group had lower FGF 19 levels, lower body weight, and lower waist circumferences among the patients with obesity and T2DM before surgery. A decreased FGF19 level might be an important risk factor for NAFLD [54]. In addition, our study suggests that the preoperative FGF 19 levels ≤ 49.97 pg/mL have a predictive role in fatty liver improvement 1 year after surgery.

## 5. Conclusions

SG effectively decreased the total bile acids levels, increased the FGF 19 levels, improved DM, and fatty liver 1 year after surgery. As the DM-CR and FL-I groups both exhibited significantly decreased total bile acid levels and increased FGF 19 levels 1 year after SG, plasma total bile acids and FGF 19 then might have some roles in DM remission and fatty liver improvement. Lower preoperative FGF 19 levels may be a predictor for fatty liver improvement after SG.

## Figures and Tables

**Figure 1 jcm-08-00815-f001:**
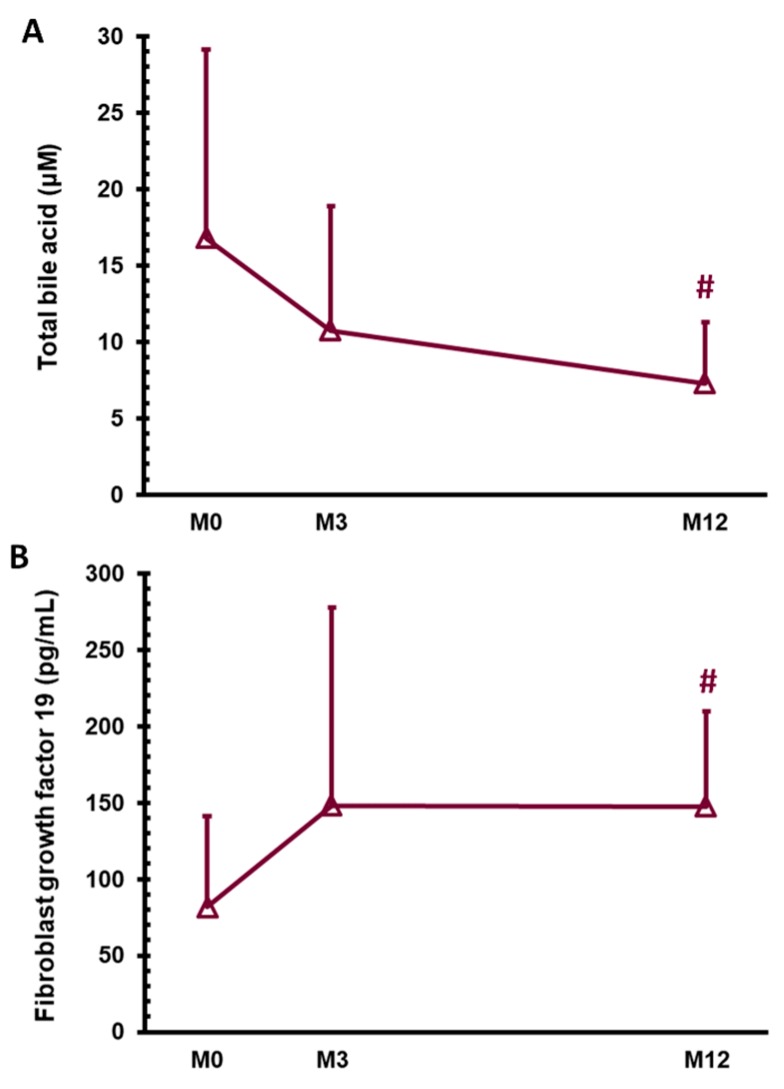
Serum levels of total bile acid (**A**) and fibroblast growth factor 19 (**B**) in patients with obesity and type 2 diabetes mellitus before (M0) and 3 months (M3) and 1 year (M12) after sleeve gastrectomy. ^#^
*p* < 0.05 compared with M0.

**Figure 2 jcm-08-00815-f002:**
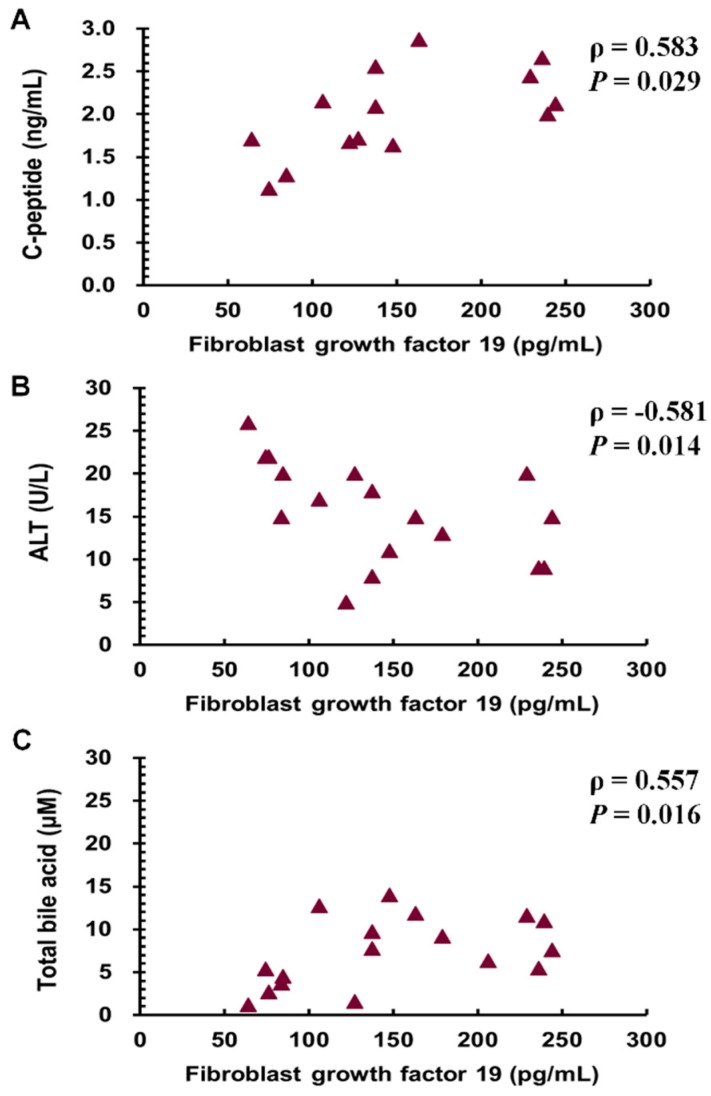
Relationships of the levels of fibroblast growth factor 19 with the levels of C-peptide (**A**), ALT (**B**), and total bile acid (**C**) in the patients with obesity and diabetes mellitus 1 year after sleeve gastrectomy. ALT, alanine aminotransferase.

**Figure 3 jcm-08-00815-f003:**
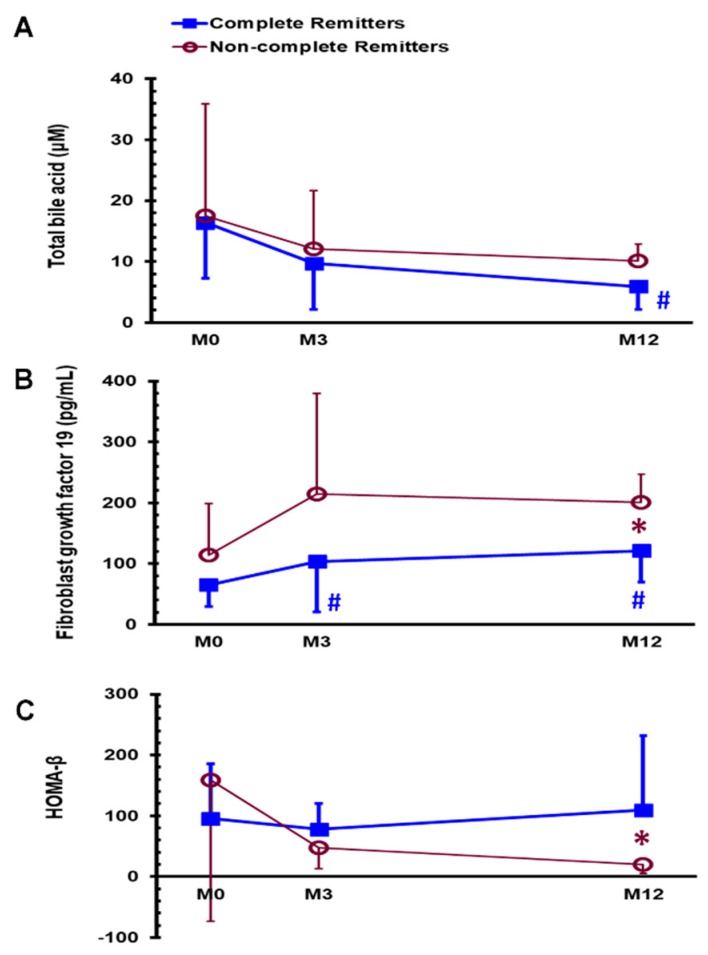
Serum levels of total bile acid (**A**) and fibroblast growth factor 19 (**B**) and HOMA-β index (**C**) in the DM-CR and DM-non-CR groups before (M0) and 3 months (M3) and 1 year (M12) after sleeve gastrectomy. * *p* < 0.05 compared between the DM-CR and DM-non-CR groups, ^#^
*p* < 0.05 compared with MDM-CR, complete remitters of diabetes mellitus; DM-non-CR, non-complete remitters of diabetes mellitus; HOMA-β, homeostasis model assessment-β.

**Figure 4 jcm-08-00815-f004:**
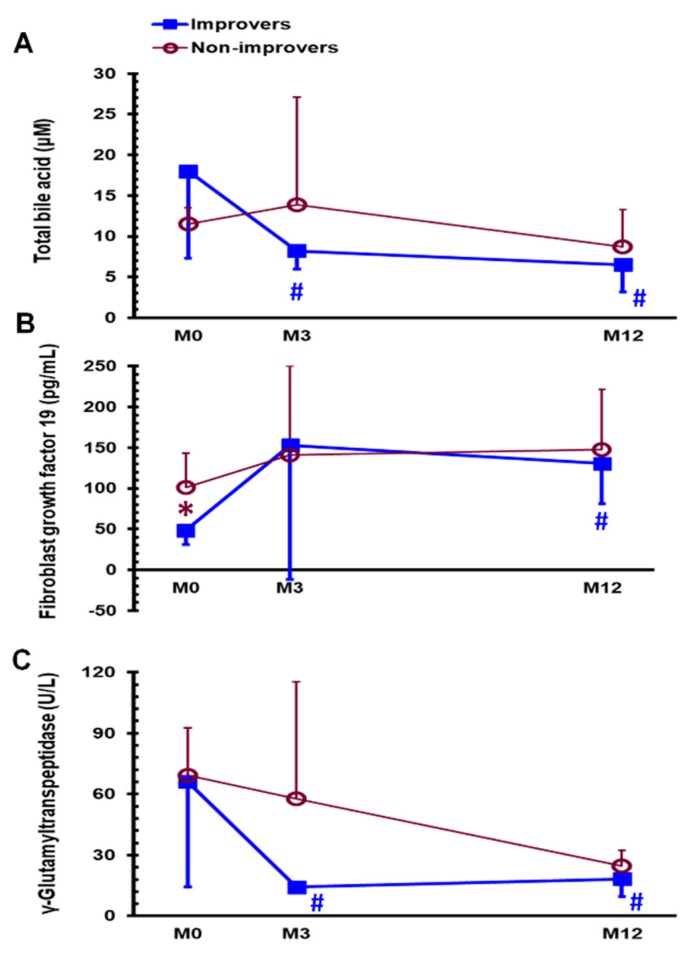
Serum levels of total bile acid (**A**) and fibroblast growth factor 19 (**B**) and γ-glutamyltranspeptidase (**C**) in the FL-I and FL-non-I groups before (M0) and 3 months (M3) and 1 year (M12) after sleeve gastrectomy. * *p* < 0.05 compared between FL-I and FL-non-I groups, ^#^
*p* < 0.05 compared with MFL-I, improvers of fatty liver; FL-non-I, non-improvers of fatty liver.

**Table 1 jcm-08-00815-t001:** Characteristics of the patients with obesity and type 2 diabetes mellitus before (M0) and 3 months (M3) and 1 year (M12) after sleeve gastrectomy.

*N* = 18	M0	M3	M12	*p* Value ^#^
Body weight (kg)	99.4 ± 19.0	79.0 ± 15.4	75.1 ± 17.0	<0.001
BMI (kg/m^2^)	35.6 ± 5.1	28.4 ± 4.3	27.0 ± 4.2	<0.001
Waist circumference (cm)	108.9 ± 11.0	92.1 ± 10.3	86.9 ± 12.7	<0.001
Excess weight loss (%)		56.4 ± 19.6	68.7 ± 20.8	<0.001
ABSI	0.078 ± 0.005	0.074 ± 0.003	0.073 ± 0.006	<0.05
Laboratory data				
Creatinine (mg/dL)	0.97 ± 0.62	1.00 ± 0.89	0.94 ± 0.55	NS
Fasting blood glucose (mg/dL)	142.1 ± 59.2	93.4 ± 8.9	91.0 ± 15.1	0.05
HbA1c (% IFCC)	8.2 ± 1.7	6.0 ± 0.5	5.8 ± 0.5	0.001
C-peptide (ng/mL)	3.8 ± 1.6	2.1 ± 1.2	2.0 ± 0.5	<0.01
Insulin (mU/L)	12.7 ± 6.5	5.3 ± 3.0	4.5 ± 2.7	0.001
Albumin (g/dL)	4.5 ± 0.3	4.4 ± 0.3	4.3 ± 0.3	<0.01
ALT (U/L)	54.9 ± 55.0	18.5 ± 9.8	15.6 ± 5.8	<0.05
AST (U/L)	34.7 ± 30.6	18.5 ± 7.8	18.1 ± 7.8	NS
Alk-p (U/L)	63.1 ± 17.0	65.3 ± 18.4	65.0 ± 20.8	NS
γ-GT (U/L)	54.0 ± 42.9	27.5 ± 34.3	19.0 ± 8.7	<0.01
Total cholesterol (mg/dL)	187.1 ± 39.5	181.0 ± 34.5	174.7 ± 29.5	NS
Triglycerides (mg/dL)	230.6 ± 147.1	106.6 ± 30.9	94.4 ± 30.0	<0.01
High density lipoprotein cholesterol (mg/dL)	37.6 ± 7.7	36.2 ± 6.6	49.9 ± 9.4	<0.01
Low density lipoprotein cholesterol (mg/dL)	116.4 ± 30.0	115.9 ± 35.6	109.2 ± 27.6	NS
Uric acid (mg/dL)	6.9 ± 1.8	5.5 ± 1.5	7.1 ± 3.0	<0.05
HOMA-IR index	4.7 ± 3.8	1.2 ± 0.7	1.0 ± 0.5	<0.001
HOMA-β index	116.9 ± 148.4	67.0 ± 41.3	74.4 ± 109.1	NS
FLI	87.1 ± 12.3	48.3 ± 25.0	29.6 ± 27.2	<0.001

All data are reported as means ± standard deviations. BMI, body mass index; ABSI, a body shape index; HbA1c, glycated hemoglobin; ALT, alanine aminotransferase; AST, aspartate transaminase; Alk-p, alkaline phosphatase; γ-GT, gamma glutamyl transferase; HOMA-IR, homeostasis model assessment, insulin resistance; HOMA-β, homeostasis model assessment-β; FLI, fatty liver index. ^#^
*p* value in the groups (Friedman’s analysis of variance).

**Table 2 jcm-08-00815-t002:** Characteristics of patients with obesity and type 2 diabetes mellitus 1 year after sleeve gastrectomy between the complete remitters of diabetes mellitus (DM-CR) and non-complete remitters of diabetes mellitus (DM-non-CR).

	DM-CR (*N* = 12)	DM-non-CR (*N* = 6)	*p* Value ^#^
Body weight (kg)	77.2 ± 18.4	71.4 ± 14.8	NS
BMI (kg/m^2^)	27.2 ± 4.3	26.5 ± 4.5	NS
Waist circumference (cm)	87.2 ± 14.2	86.3 ± 10.0	NS
ABSI	0.074 ± 0.004	0.073 ± 0.009	NS
Lab data			
Creatinine (mg/dL)	0.81 ± 0.18	1.21 ± 0.96	NS
Fasting blood glucose (mg/dL)	85.0 ± 8.1	102.0 ± 19.4	NS
HbA1c (% IFCC)	5.5 ± 0.2	6.3 ± 0.5	<0.001
C-peptide (ng/mL)	1.9 ± 0.5	2.2 ± 0.4	NS
Insulin (mU/L)	5.4 ± 2.7	2.4 ± 0.7	<0.05
Albumin (g/dL)	4.3 ± 0.2	4.4 ± 0.4	NS
ALT (U/L)	16.8 ± 5.9	12.6 ± 4.9	NS
AST (U/L)	18.3 ± 7.7	17.8 ± 9.5	NS
Alk-p (U/L)	60.5 ± 18.1	77.5 ± 25.3	NS
γ-GT (U/L)	18.1 ± 7.7	21.5 ± 11.8	NS
FGF 19 (pg/mL)	121.1 ± 51.4	200.6 ± 46.9	<0.05
Total bile Acid (μM)	5.9 ± 3.8	10.1 ± 2.8	NS
Total cholesterol (mg/dL)	177.1 ± 30.4	169.4 ± 29.8	NS
Triglycerides (mg/dL)	88.5 ± 33.2	107.4 ± 17.5	NS
High density lipoprotein cholesterol (mg/dL)	52.2 ± 9.1	44.3 ± 8.8	NS
Low density lipoprotein cholesterol (mg/dL)	109.2 ± 28.5	109.2 ± 28.9	NS
Uric acid (mg/dL)	6.9 ± 3.5	7.3 ± 2.1	NS
HOMA-IR index	1.13 ± 0.56	0.63 ± 0.22	NS
HOMA-β index	109.4 ± 122.1	20.4 ± 15.0	0.010
FLI	31.0 ± 31.2	24.9 ± 5.5	NS

All data are reported as means ± standard deviations. BMI, body mass index; ABSI, a body shape index; HbA1c, glycated hemoglobin; ALT, alanine aminotransferase; AST, aspartate transaminase; Alk-p, alkaline phosphatase; γ-GT, gamma-glutamyl transferase; FGF, fibroblast growth factor; HOMA-IR, homeostasis model assessment, insulin resistance; HOMA-β, homeostasis model assessment-β; FLI, fatty liver index. ^#^
*p* value in groups (Friedman’s analysis of variance).

**Table 3 jcm-08-00815-t003:** Characteristics of patients with obesity and type 2 diabetes mellitus 1 year after sleeve gastrectomy between the improvers of fatty liver (FL-I) and non-improvers of fatty liver (FL-non-I) based on the FLI.

	FL-I (*N* = 9)	FL-non-I (*N* = 4)	*p* Value ^#^
Body weight (kg)	69.0 ± 8.0	98.6 ± 10.8	<0.01
BMI (kg/m^2^)	24.8 ± 1.8	32.1 ± 3.1	<0.01
Waist circumference (cm)	80.8 ± 4.4	97.8 ± 18.3	NS
ABSI	0.074 ± 0.004	0.073 ± 0.009	NS
Laboratory data			
Creatinine (mg/dL)	0.87 ± 0.19	0.76 ± 0.12	NS
Fasting blood glucose (mg/dL)	94.1 ± 16.7	84.0 ± 16.4	NS
HbA1c (% IFCC)	5.8 ± 0.5	5.7 ± 0.3	NS
C-peptide (ng/mL)	1.8 ± 0.5	2.3 ± 0.4	NS
Insulin (mU/L)	3.7 ± 2.4	6.7 ± 2.2	NS
Albumin (g/dL)	4.4 ± 0.3	4.2 ± 0.2	NS
ALT (U/L)	15.7 ± 5.9	17.3 ± 3.3	NS
AST (U/L)	18.9 ± 7.0	13.5 ± 1.0	NS
Alk-p (U/L)	67.8 ± 17.0	62.0 ± 28.8	NS
γ-GT (U/L)	18.3 ± 8.7	24.8 ± 7.5	NS
FGF 19 (pg/mL)	130.5 ± 49.6	147.4 ± 73.9	NS
Total bile Acid (μM)	6.5 ± 3.3	8.7 ± 4.6	NS
Total cholesterol (mg/dL)	181.1 ± 27.1	180.8 ± 34.7	NS
Triglycerides (mg/dL)	83.8 ± 24.9	123.3 ± 33.0	0.05
High density lipoprotein cholesterol (mg/dL)	55.9 ± 6.2	38.5 ± 5.3	<0.05
Low density lipoprotein cholesterol (mg/dL)	110.6 ± 26.3	125.8 ± 26.9	NS
Uric acid (mg/dL)	7.9 ± 3.5	5.6 ± 1.6	NS
HOMA-IR index	0.82 ± 0.47	1.34 ± 0.36	NS
HOMA-β index	42.6 ± 41.2	204.1 ± 180.2	0.05
FLI	15.1 ± 6.8	62.3 ± 28.1	<0.01

All data are reported as means ± standard deviations. BMI, body mass index; ABSI, a body shape index; HbA1c, glycated hemoglobin; ALT, alanine aminotransferase; AST, aspartate transaminase; Alk-p, alkaline phosphatase; γ-GT, gamma-glutamyl transferase; FGF, fibroblast growth factor; HOMA-IR, homeostasis model assessment, insulin resistance; HOMA-β, homeostasis model assessment-β; FLI, fatty liver index. ^#^
*p* value in groups (Friedman’s analysis of variance).

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
