# Peer review of "Bile Acid and Fibroblast Growth Factor 19 Regulation in Obese Diabetics, and Non-Alcoholic Fatty Liver Disease after Sleeve Gastrectomy"

_jcm, 2019, doi:10.3390/jcm8060815_

Round 1
Reviewer 1 Report
The manuscript was reviewed for publication in the journal. The manuscript was designed to investigate the roles of total bile acids and FGF 19 in T2DM remission and NAFLD improvement in obese subjects undergoing sleeve gastrectomy. The results obtained showed that sleeve gastrectomy significantly decreased the total bile acids levels, increased the FGF 19 levels, induced DM remission, and decreased the fatty liver improvement 1 year after surgery. It is the reviewer’s opinion that the manuscript is interesting and easy to follow. However, it appears that there are a couple of major concerns in the manuscript.
1) There are many papers regarding the role of bile acids after sleeve gastrectomy and a couple of papers about the relationship between FGF 19 and sleeve gastrectomy. The authors should discuss about the related papers and show what is new in the paper compared with the previous papers.
2) The authors should use fatty liver index (FLI) to be defined as NAFLD. There are many non-invasive markers for NAFLD. The authors should express why the authors choose FLI.
3) The paper indicated that a total of 18 patients were enrolled in this study; however, table 3 showed only 13 patients of the improvers of fatty liver (N=9) and non-improvers of fatty liver (N=4). What are the 5 patients excluded? These patients may show normal FLI before sleeve gastrectomy? The authors should explain the issue.
4) The study clearly showed that sleeve gastrectomy significantly decreased the total bile acids levels, increased the FGF 19 levels, induced DM remission, and decreased the fatty liver improvement. However, it is not clear that the changes in bile acids levels and FGF 19 levels may be a result or cause of sleeve gastrectomy. The authors should discuss about the issue.
5) Table 2 and 3 showed the characteristics of patients with obesity and type 2 diabetes mellitus 1 year after sleeve gastrectomy between the improvers and non-improvers of diabetes mellitus and fatty liver. The authors should show the characteristics of these patients before sleeve gastrectomy between the improvers and non-improvers of diabetes mellitus and fatty liver.
Author Response
Reviewer #1 Evaluation
We appreciate for your suggestions about our current work, and we have modified our literature and made specific responses to your opinions as the followings:
1. There are many papers regarding the role of bile acids after sleeve gastrectomy and a couple of papers about the relationship between FGF 19 and sleeve gastrectomy. The authors should discuss about the related papers and show what is new in the paper compared with the previous papers.
Answer: We added the description to emphasize the new in our article (Discussion section: Page 12, lines 45-46; Page 13, lines 46-51). Thank you for the suggestion.
2. The authors should use fatty liver index (FLI) to be defined as NAFLD. There are many non-invasive markers for NAFLD. The authors should express why the authors choose FLI.
Answer: We used another two non-invasive scores for NAFLD and cited Ref 24. & Ref 25. We added “We also applied hepatic steatosis index [24] and lipid accumulation product [21, 25] to verify our improvement of NAFLD one year after SG by FLI. SG successfully decreased the hepatic steatosis index and lipid accumulation product scores one year after SG (p < 0.001).” (Results section: Page 4, lines 25-28). Thank you for the suggestion.
3. The paper indicated that a total of 18 patients were enrolled in this study; however, table 3 showed only 13 patients of the improvers of fatty liver (N=9) and non-improvers of fatty liver (N=4). What are the 5 patients excluded? These patients may show normal FLI before sleeve gastrectomy? The authors should explain the issue.
Answer: We added the description “One year after SG, 5 patients were excluded due to missing data at 1 year after SG in the calculation of FLI (2 patients without triglyceride level, 2 patients without gamma glutamyl transferase, and 1 patient without waist circumference). ” (Results section: Page 9, line 10-12). The FLI of the excluded 5 patients before SG were 73.6, 79.8, 57.7, 98.2, and 82.2. All of them were more than 30 (normal FLI<30). Thank you.
4. The study clearly showed that sleeve gastrectomy significantly decreased the total bile acids levels, increased the FGF 19 levels, induced DM remission, and decreased the fatty liver improvement. However, it is not clear that the changes in bile acids levels and FGF 19 levels may be a result or cause of sleeve gastrectomy. The authors should discuss about the issue.
Answer: We added “FXR-knockout mice have been shown to lose the abilities of reducing body weight and improving glucose tolerance after SG. The results of our study found that high total bile acid, probably due to dysregulation of the FXR in T2DM, and subsequently low FGF 19 were detected before SG” (Discussion section: Page 12, lines 38-41).
5. Table 2 and 3 showed the characteristics of patients with obesity and type 2 diabetes mellitus 1 year after sleeve gastrectomy between the improvers and non-improvers of diabetes mellitus and fatty liver. The authors should show the characteristics of these patients before sleeve gastrectomy between the improvers and non-improvers of diabetes mellitus and fatty liver.
Answer: We added “In Table 2, there was no any statistical significant difference between complete remitters of DM (DM-CR group) and the non-complete remitters of DM (DM-non-CR group) in any variables before SG” (Results section, Page 7, lines 8-10). We also added “Before SG, the improvers of fatty liver (FL-I group) demonstrated significantly lower body weight (94.4.0 ± 9.7 vs. 121.3 ± 19.7 kg, p < 0.05), lower waist circumferences (104.7 ± 6.1 vs. 122.9 ± 12.0 kg/m2, p < 0.05), lower FGF 19 (48.7 ± 18.0 vs. 101.2 ± 42.4%, p < 0.05) than the non-improvers of fatty liver (FL-non-I group)” (Results section, Page 9, lines 7-10). Thank you very much for suggestion.
Reviewer 2 Report
The authors present a cohort study of 18 patients undergoing laparoscopic sleeve gastrectomy for moderate obesity with type 2 diabetes mellitus and NAFLD. Underlying mechanisms concerning improvement of NAFLD and T2DM are evaluated with regard to bile acids and FGF19. The topic is of interest. The manuscript needs the following revisions:
Some language editing by a native speaker should be performed.
"A standard 5-trocar laparoscopic technique was applied using a laparoscopic stapler (EndoGIA; Tyco, United States Surgical Corporation, Norwalk, CT, USA) with a 60-cm cartridge (3.5-mm stapler height, blue load) to divide the stomach ~2 cm wide along the lesser curvature side."
I believe the authors used a 60-mm cartridge rather than 60-cm?
The following studies and reviews have evaluated different procedures for metabolic surgery, ie treatment of NAFLD and T2DM and should be discussed along with the findings from the present manuscript:
MANAGEMENT OF ENDOCRINE DISEASE: Which metabolic procedure? Comparing outcomes in sleeve gastrectomy and Roux-en Y gastric bypass
AT Billeter, JR de la Garza Herrera, KM Scheurlen, F Nickel, F Billmann, ...
European journal of endocrinology 179 (2), R77-R93
Bariatric Surgery as an Efficient Treatment for Non-Alcoholic Fatty Liver Disease in a Prospective Study with 1-Year Follow-up
F Nickel, C Tapking, L Benner, J Sollors, AT Billeter, HG Kenngott, ...
Obesity surgery 28 (5), 1342-1350
In the latter study several NAFLD scores were used together with transient elastography for evaluation of NAFLD. The authors of the present manuscript should also at least use NAFLD fibrosis score in addition to FLI in their study to enable comparability to other studies.
Conclusions:
"SG significantly decreased the total bile acids levels, increased the FGF 19 levels, induced DM
remission, and decreased the FLI 1 year after surgery."
This statement should be toned down since not all parameters were improved in all patients.
"As the DM-CR and FL-I groups both exhibited significantly decreased total bile acid levels and increased FGF 19 levels 1 year after SG, plasma total bile acids and FGF 19 then play pivotal roles in DM remission and fatty liver improvement. Lower preoperative FGF 19 levels are a predictor for fatty liver improvement after SG."
These statements should be toned down since the evidence from the present study is preliminary and limited.
Author Response
Reviewer #2 Evaluation
1. Some language editing by a native speaker should be performed.
Answer: This article have been edited by a professional native speaker. Thank you very much.
2. "A standard 5-trocar laparoscopic technique was applied using a laparoscopic stapler (EndoGIA; Tyco, United States Surgical Corporation, Norwalk, CT, USA) with a 60-cm cartridge (3.5-mm stapler height, blue load) to divide the stomach ~2 cm wide along the lesser curvature side."
I believe the authors used a 60-mm cartridge rather than 60-cm?
Answer: We revised it and thanks for your correction (Method section: Page 3, line 17).
3. The following studies and reviews have evaluated different procedures for metabolic surgery, ie treatment of NAFLD and T2DM and should be discussed along with the findings from the present manuscript:
“MANAGEMENT OF ENDOCRINE DISEASE: Which metabolic procedure? Comparing outcomes in sleeve gastrectomy and Roux-en Y gastric bypass
AT Billeter, JR de la Garza Herrera, KM Scheurlen, F Nickel, F Billmann, ...
European journal of endocrinology 179 (2), R77-R93
Bariatric Surgery as an Efficient Treatment for Non-Alcoholic Fatty Liver Disease in a Prospective Study with 1-Year Follow-up
F Nickel, C Tapking, L Benner, J Sollors, AT Billeter, HG Kenngott, ...
Obesity surgery 28 (5), 1342-1350
In the latter study several NAFLD scores were used together with transient elastography for evaluation of NAFLD. The authors of the present manuscript should also at least use NAFLD fibrosis score in addition to FLI in their study to enable comparability to other studies.
Answer: We added the statement about the different procedures for metabolic surgery on the treatment of NAFLD and T2DM (Page 13, lines 39-45), cited the suggested papers with Ref. 56 and 57. Owing to the limited data, we cannot use NAFLD fibrosis score to analyze the improvement of NAFLD. In addition to FLI, we further utilized hepatic steatosis index and lipid accumulation product to verify the improvement of NAFLD after SG in our study. The added description is “We also applied hepatic steatosis index [24] and lipid accumulation product [21, 25] to verify our improvement of NAFLD one year after SG by FLI. SG successfully decreased the hepatic steatosis index and lipid accumulation product scores one year after SG (p < 0.001)” (Results section: Page 4, lines 25-28).
Transient elastography is also a non-invasive measurement and often used to measure the liver stiffness. We plan to conduct another study to investigate the improvement of NAFLD on the patients with obesity and T2DM by FLI score, NAFLD fibrosis score, and transient elastography. Thank you very much for suggestion.
4. Conclusions:
"SG significantly decreased the total bile acids levels, increased the FGF 19 levels, induced DM remission, and decreased the FLI 1 year after surgery."
This statement should be toned down since not all parameters were improved in all patients.
Answer: We toned down the description and revised to “SG effectively decreased the total bile acids levels, increased the FGF 19 levels, improved DM and fatty liver 1 year after surgery.” (Conclusion section: Page 14, lines 1-2). Thank you.
5. "As the DM-CR and FL-I groups both exhibited significantly decreased total bile acid levels and increased FGF 19 levels 1 year after SG, plasma total bile acids and FGF 19 then play pivotal roles in DM remission and fatty liver improvement. Lower preoperative FGF 19 levels are a predictor for fatty liver improvement after SG."
These statements should be toned down since the evidence from the present study is preliminary and limited.
Answer: We toned down the description and revised to “As the DM-CR and FL-I groups both exhibited significantly decreased total bile acid levels and increased FGF 19 levels 1 year after SG, plasma total bile acids and FGF 19 then might have some roles in DM remission and fatty liver improvement. Lower preoperative FGF 19 levels may be a predictor for fatty liver improvement after SG.” (Conclusion section: Page 14, lines 2-6). Thank you.
Reviewer 3 Report
This research paper turns the light on the relationship between plasma bile acids and FGF 19 in diabetes remission and NAFLD improvement aftet Sleeve gastrectomy.
However, several promising studies are discussing today the bile acids signaling as a pharmacological therapeutic approach to treat obesity and its metabolic complications mainly diabetes mellitus.
In this paper some points should be addressed:
1) In the text there is a lack of data regarding the anti-diabetic drugs, Insulin-Therapy, GLP-agonists and statins at baseline. This is essential in influencing the response at M3 as well at M12 in terms of diabetes remission and NAFLD improvement in patients subjected to bariatric surgery.
Moreover, did all diabetic patients reached HbA1c reduction post-sleeve under same antidiabetic treatment?
2) Today life style- based Interventions is considered the gold standard for the improvement of NAFLD and involved crucially in improving insulin resistance. Did the patients involved in the study were subjected to similar life style interventions during the follow-up period? It is worthy to highlight this point and describe its intensity. Life style interventions could be a strong confounder involved in modification of bile acids concentration and consequently FGF 19 levels
3) Data from literature have already shown an increase in FGF 19 and bile acids following Roux-en-Y Gastric Bypass. Therefore, which advantages did Sleeve gastrectomy in terms of plasma levels as ell as physiology of bile acids and expression of FGF 15 could offer in comparison to other bariatric interventions (e.g. Roux-en-Y)
4) It is worth to explain how low preoperative FGF 19 levels are a predictor for NAFLD improvement after SG by adding possibly supporting data from literature.
Author Response
Reviewer #3 Evaluation
1. In the text there is a lack of data regarding the anti-diabetic drugs, Insulin-Therapy, GLP-agonists and statins at baseline. This is essential in influencing the response at M3 as well at M12 in terms of diabetes remission and NAFLD improvement in patients subjected to bariatric surgery.
Moreover, did all diabetic patients reached HbA1c reduction post-sleeve under same antidiabetic treatment?
Answer: We added the descriptions “Before SG, the medications for treating DM were oral anti-diabetic drugs in 16 patients (16/18, 88.9%), insulin in 2 patients (2/18, 11.1%), and statins in 8 (8/18, 44.4%) patients. After SG, only 1 (1/18, 5.6%) patient used oral anti-diabetic drugs 3 months after surgery, and 1 (1/18, 5.6%) patient used statins 1 year after surgery. None of the patients ever used glucagon-like peptide-1 agonist.” (Results section: Page 7, lines 4-8) Thank you very much.
2. Today life style- based Interventions is considered the gold standard for the improvement of NAFLD and involved crucially in improving insulin resistance. Did the patients involved in the study were subjected to similar life style interventions during the follow-up period? It is worthy to highlight this point and describe its intensity. Life style interventions could be a strong confounder involved in modification of bile acids concentration and consequently FGF 19 levels
Answer: We added the description “After SG, all patients received the clinics follow-up and consultation of dieticians. All were encouraged life style modifications, such as walking 10,000 steps/per day and strict diet control.” (Method section: Page 3, lines 11-12). Thanks for your suggestion.
3. Data from literature have already shown an increase in FGF 19 and bile acids following Roux-en-Y Gastric Bypass. Therefore, which advantages did Sleeve gastrectomy in terms of plasma levels as well as physiology of bile acids and expression of FGF 15 could offer in comparison to other bariatric interventions (e.g. Roux-en-Y)
Answer: At present, the changes of bile acids on patients with obesity and DM after RYGB are inconsistent, they may decrease [Gerhard GS 2013 Diabetes Care], or increase [Nemati R 2018 Obes Surg]. Also the same condition occurs in the patients with obesity and DM after SG. We showed that “Previous studies on total bile acid levels after SG reported inconsistent finding; they may increase [28], not change [14, 33] or decrease [34]” (Discussion section: Page 12, lines 35-36). In different metabolic surgery, SG had a better capability to normalize liver function test than Roux-en-Y gastric bypass in patients with obesity and T2DM 1 year after surgery [56]. (Discussion section: Page 13, lines 40-42). And we added new description “Owing to the Y-Y paradox, i.e. identical BMI with different body fat in different ethnic populations [37], our previous study also showed that the 2-year rate of DM-CR was only 5.5% after SG, much less than 50% after gastric bypass [38]. The new findings in our study suggest the close relationship between total bile acid and DM remission after SG in our Asian population. The regulation of FXR might play an important role in subjects with obesity and DM”. (Discussion section, Page 12, line 50 to Page 13, line 3) Thanks for your suggestion.
4. It is worth to explain how low preoperative FGF 19 levels are a predictor for NAFLD improvement after SG by adding possibly supporting data from literature.
Answer: We added the description “A receiver operating characteristic (ROC) curve analysis was performed to determine the potential role of FGF 19 level in the discrimination of FL-I group and FL-non-I group. The area under ROC curve between the two groups was 0.917 (95% confidence interval: 0.631-0.997; p < 0.0001). The optimal cut-off value for FGF 19 level between the two groups was 49.97 pg/mL. When FGF 19 was ≤49.97 pg/mL, the sensitivity, specificity, positive predictive value and negative predictive value for FL-I group were 77.8%, 100.0%, 100% and 66.7%, respectively.” (Results section: Page 9, line 22 to Page 10 line 2). ROC curve calculation is from MedCalc statistical software, and the figure of ROC is shown be below (see attached word file). Thanks for your suggestion.
Reviewer 4 Report
This is a small clinical observational study indicating relative relevance of increased FGF19 post SG as a surgical treatment for metabolic disorder and obesity.
The milieu post surgery provides an important observational opportunity to identify novel biomarkers or factors that may contribute to the phenotype. FGF19 increase and total bile acid decrease indicate co-occurring, sustained and the inverse relationship between the two and apparently beneficial effect on NAFLD. FGF19 increase further separated the complete vs non-complete responders to GS, probably via TGF5 axis. In summary, this is a good observational study but without much mechanistic insights.
Author Response
Reviewer #4 Evaluation
1. This is a small clinical observational study indicating relative relevance of increased FGF19 post SG as a surgical treatment for metabolic disorder and obesity.
The milieu post surgery provides an important observational opportunity to identify novel biomarkers or factors that may contribute to the phenotype. FGF19 increase and total bile acid decrease indicate co-occurring, sustained and the inverse relationship between the two and apparently beneficial effect on NAFLD. FGF19 increase further separated the complete vs non-complete responders to SG, probably via TGF5 axis. In summary, this is a good observational study but without much mechanistic insights.
Answer: Activation of the bile acid-responsive G protein-coupled receptor TGR5 is known to increase energy expenditure and reduce fat mass by increasing basal metabolic rate. The bile acid–TGR5 axis as a novel pathway that elicits beige remodelling in subcutaneous white adipose tissue and modulates mitochondrial function in mice. We plan to conduct another study to investigate the relationship of bile acid, FGF 19 and TGR5 between complete vs. non-complete responders of NAFLD on the patients with obesity and T2DM. Thank you very much.
Round 2
Reviewer 1 Report
The manuscript was re-reviewed for publication in the journal. It is the reviewer’s opinion that the manuscript was properly revised based on the reviewer’s comments. I have no comment in the manuscript.
Reviewer 2 Report
Authors addressed the comments. English language should be improved.